# Resource Efficient Deep Reinforcement Learning for Acutely Constrained TinyML Devices

## ABSTRACT

The use of Deep Reinforcement Learning (Deep RL) in many resource constrained mobile systems has been limited in scope due to the severe resource consumption (e.g., memory, computation, energy) such approaches exert. As a result, TinyML devices ranging from sensors and cameras to small form-factor robots and drones have been unable to benefit from the advantages of recent Deep RL algorithms that have underpinned breakthrough results in applications of decision and control. In this work, we propose and study a variety of general-purpose techniques designed to lower such system resource bottlenecks for Deep RL by optimizing both the agent algorithms and neural architectures used in these solutions. Experiments show our Deep RL optimization framework that combines these techniques is able produce significant efficiency gains to the point such techniques become feasible for TinyML platforms. We present one representative end-to-end application (viz. network protocol learning) executing on constrained processors (embedded-hardware), in addition to simulated control problems addressed assuming limited access to system resources.

## 1. INTRODUCTION

In the past five years deep learning has made important strides within critical, typically discriminative, learning tasks. But beyond these sit an additional set of breakthroughs that have been achieved within the area of reinforcement learning powered largely again through the use of deep neural network principles and algorithms; referred to as Deep Reinforcement Learning (Deep RL). Many of the more celebrated results within the popular press center around the ability of such approaches to train algorithms that reach human-level performance across a wide-range of games of skill; for example, playing 50 different Atari games or in games that have long resisted automated cognition such as AlphaGo [30] and DeepStack [23] – the games of Go and poker respectively.

Deep RL, offers the ability to learn algorithms that make decision on the basis of inputs (such as sensors or system state). Potential (and emerging) mobile applications of such techniques include: determining when and how mobile system might schedule and regulate resources; or driving the decisions of a drone to navigate a flight path. But *deep*-forms of reinforcement learning in particular have not yet extensively been applied to a variety of these tasks performed by the more resource constrained forms of mobile devices and robots drones;

the reason for this is because of the extreme system resources Deep RL requires. Most of the examples of systems applying these techniques have enormous resource budgets; for instance, the control decisions of autonomous cars, of course, already have adopted Deep RL techniques [8]. But platforms like the Nvidia Drive PX2 [2] designed for such scenarios require 250W of power while offering 8 tera-flops of computation. As an extreme example, for AlphaGo to beat the world's best human player required more than 1,200 CPUs and 176 GPUs [30]. However such resource levels are out of reach for embedded systems ranging from smart sensors to small form-factor drones and robots. To bring Deep RL to these devices will require a leap in the efficiency, but today what we lack is a principled understanding of how to reduce and manage the resources used by deep reinforcement learning algorithms. It is unclear for example, how well these algorithms would perform when potential avenues of optimization are explored – and if acceptable control and decision performance could be maintained within the constraints of tinyML devices. Only recently has this area begun to be explored in the literature [11, 33, 19, 28].

In this work, we perform an early exploration towards general-purpose system resource optimizations techniques for deep reinforcement learning (Deep RL) algorithms. Our aim is to enable this class of algorithms to execute specifically on severely limited processors such as those found in tinyML platforms like small form-factor IoT devices, drones and robots – with typical processors that can be as limited as ARM Cortex M0 processors. To achieve this, we study methods to reduce the memory, computation and energy requirements of Deep RL algorithms largely at the expense of a manageable loss in algorithm performance (i.e., the ability to first learn, and then perform control and decision tasks). We build where possible on growing knowledge of optimization techniques for deep learning classification models at inference time, such use of low-precision parameters, separation of convolution layers and resource-sensitive model selection (e.g., [5]). Although direct application of such methods designed for discriminative deep models is limited by a series of intrinsic differences with Deep RL models that presents a mixture of unique resource bottlenecks, along with differing algorithmic data-flows and neural network architectures. Unlike deep classifiers that are trained off-line before they are introduced to a constrained device, Deep RL often revises model

parameters iteratively based on the arrival of new data; as a result, not only must the forward-pass of the neural networks used in DRL be optimized, but also those for back-propagation. Similarly, although DeepRL often use similar neural architectures, that typically act as perceptional phases applied to incoming data – these networks may only consume a fraction of the overall resources used.

The contributions of this work are:

- **Early Mobile Insights into Deep RL.** As far as we are aware, our results demonstrating a range of Deep RL tasks on the severely constrained Cortex M0 are some of the first-of-their-kind. We show the feasibility for reasonable trade-offs of algorithm performance (i.e., reward function) and system resources (like memory) which opens the door to many new powerful Deep RL applications.

- **Systematic Low-resource Deep RL Study.** We devise and investigate the general purpose use of a range of techniques: reduced precision, agent and architecture selection, and redundancy minimization. When collectively applied these methods lower the resource needs of Deep RL to the point it can locally run on embedded hardware; significantly, none of these techniques are tied to specific RL tasks.

- **TinyML Deep RL Application Evaluation.** We study a distinct tinyML application: a distributed co-ordination task in Deep RL that performs a custom message-passing protocol for a specific topology and environment. For the first time, our methods show this recent theoretical result [12] can execute even on a network of Cortex M0 nodes.

## 2. CHALLENGES OF MOBILE DEEP RL

In this section, we briefly provide background into deep reinforcement learning theory and its potential to be applied to embedded and mobile systems.

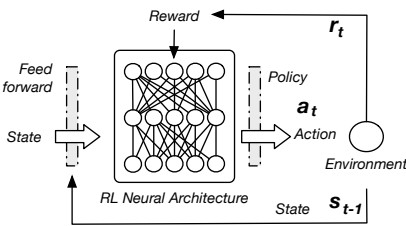

**Figure 1:** Reinforcement learning cycle.

### 2.1 Deep RL Primer

Intuitively, learning can be understood as the process of interacting with an environment which as a result forms experiences. Based on these experiences with the surroundings, implicit connections in terms of cause and

effect can be formed. The desired actions and consequences to obtain and achieve a designated objective in this environment can be mapped out.

In many Reinforcement Learning applications the scenario can be framed more formally with the main elements of the *agent*, a *policy*, a *reward*, *value function* and the *environment* and in certain scenarios a *model* of the environment. The interactions of each element in the reinforcement learning setting are illustrated in Figure 1 shown below.

Where, $a_t \in A$ (set of all actions), $s_t \in S$ (set of all states), $r_t \in R$ (set of all rewards). Thus, through reinforcement learning the process of learning is performed by the agent interacting with the environment and reinforcing positive or negative behaviour to obtain an objective, or noted as goal-directed learning.

### 2.2 TinyML Deep RL Scenarios

The ability to overcome the limitations of Deep RL-style algorithms with general-purpose optimization techniques for resource constrained devices would be broadly beneficial to a host of exciting learning tasks such as robots, drones and other sensor based networks. The optimization techniques proposed should be easily executed on low-resource platforms, minimize DRL model changes and generalize well enough for a variety of possible applications. For example, sensor networks with global/local co-ordination [21], robust low complexity control [24] and real-time plane management [31],[32]. The application to Deep RL in drone swarming [26] setting presents another possible course of action.

## 3. DEEP RL EFFICIENCY FRAMEWORK

To enable control, and more importantly reduction, of system resources consumed by Deep RL we developed an efficiency framework illustrated in Figure 2. The heart of this framework is a series of optimization techniques detailed in §4.

### 3.1 Framework Architecture

Our efficiency framework is comprises by the following four core artefacts shown in Figure 2.

**Deep RL Module.** Initially the user of the framework will specify a preliminary Deep RL module. This module is otherwise unremarkable and so includes internals such as an agent algorithm, neural architecture, state update and forward propagation operations. This module is the target of the framework's efforts to reduce the resource consumption. It can be replaced with another RL module once the user is satisfied with the system resource performance achieved.

**Resource Scalers.** An extensible number of techniques are included that operate on the neural architecture, state update and forward propagation operations of the

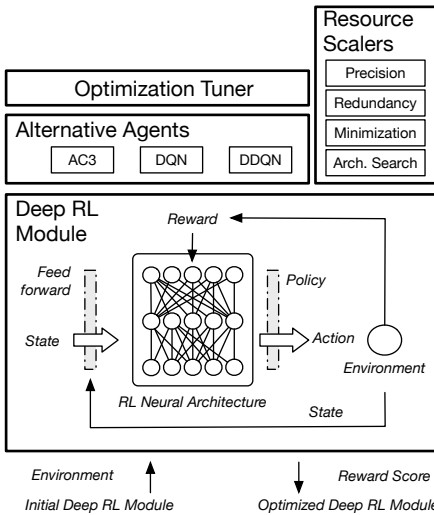

**Figure 2:** DQN Optimisation Process.

Deep RL module. Each technique has a different capability to shape one or more resources (e.g., memory and energy), by typically approximating its normal operation. We find in most cases that significant amounts of resource usage can be reduced for only a small loss of quality of it behavior. At this stage the framework includes the following individual scalers: (1) selective mixed parameter precision; (2) architecture search; (3) redundancy minimization; (4) convolution optimization.

**Optimization Tuner.** The previous framework components provide a variety of options for shaping the resource consumption. The role of the Optimization Tuner is to determine how these options should be combined, and where applicable, to what level of intensity they should be applied. The aim is not for this component to arrive the absolute optimal mixture of settings – only for this tuning to be simple and tractable.

## 3.2 Framework Operation

Framework initiation begins with the provision of a Deep RL module along with an environment (either real or simulated). The iterative execution of the RL module within the environment is started and this guides the Optimization Tuner as to how heavily to pursue opportunities to reduce resources is given by the *Retain Rate.* This value attempts to act as a universal summary of optimization intensity across all available optimization methods. Feedback to the Optimization Tuner is provided by the Reward Score computed by the Deep RL module which is necessary part of any RL formulation that indicates the quality of decisions being made within the context of the task being performed.

The Optimization Tuner operates iteratively with the provided environment feedback given to the current spec-

ification of the Deep RL module. The Optimization Tuner can explore potential optimizations by explicitly turning on and off certain techniques or changing the intensity at which Resource Scalers are applied.

Primarily, the Resource Scalers will make modifications to the Neural Architecture of the Deep RL Module. For example, separating a convolutional layer to lower memory or compute demands. But optimizations are also applied to the operations necessary to revise policy decisions, which open up new potential inefficiencies that occur during the updating of the Neural Architecture.

## 4. DEEP RL EFFICIENCY TECHNIQUES

We now specify in more detail each major component of the framework described in the prior section. We begin with each of the four Resource Scalers, before ending with the Optimization Tuner.

### 4.1 Selective Mixed Parameter Precision

For deep models designed for classification tasks (e.g., recognize a face), it well-established that the model accuracy can remain surprisingly high even if the parameters of the model are changed from 32-bits to 8-bits or even less [14, 10]. The theoretical reasons for this are not yet well understood, but it is reasoned that the ability of deep learning to tolerate noisy complex inputs is similar to withstanding a loss of precision in how the data is represented within the model. However, this *property of neural networks has not been explored significantly within Deep RL*, even though clearly it is reasonable to anticipate it might.

Motivated by this, within our optimization framework we incorporate *stochastic rounding* [14]. Under this approach, any Deep RL parameter can be modified to a certain precision level simply by using:

$$Round(x, \langle \mathbb{IL}, \mathbb{FL} \rangle) = \begin{cases} \lfloor x \rfloor & w.p \quad 1 - \frac{x - \lfloor x \rfloor}{\epsilon} \\ \lfloor x \rfloor + \epsilon & w.p \quad x + \frac{x - \lfloor x \rfloor}{\epsilon} \end{cases}$$

Where, $\mathbb{IL}$ refers to the number of integer bits and $\mathbb{FL}$ refers to the number of fractional bits. To regulate and select which parameters are simplified a greedy search is performed with parameters modified in batches of layers or block of matrices for non-neural network architecture parameters. Architecture parameters are prioritized ahead of all others. A user can set the retain value to indirectly determine the fraction of parameters that are modified and by how much.

### 4.2 Redundancy Minimization

Significant inefficiencies exist within the neural architecture of a Deep RL module. The following matrix factorization approach has been used for classification tasks in deep learning previously [34, 20] *but as far as we are*

*aware never within a Deep RL module, and therefore it remained unknown just how effective this approach would turn out to be.* More over, we also apply such matrix factorization based efficiency gains to Deep RL specific operations (e.g., updating policy gradients).

Formally, the process is as follows: $\mathbf{x}^L \in \mathbb{R}^{m \times n}$ represents the states of the nodes in the previous layer. All connections between layer $L$ and layer $L+1$ containing $m$ and $n$ units respectively can be represented by the matrix, $\mathbf{W}^L$. Let the $i^{th}$ row of $\mathbf{W}^L$ contain all the connections between the $i_{th}$ unit in layer $L$ and all the units in layer $L+1$. Thus, the redundancy reduction of the fully connected layer in a neural network can be achieved by the well known matrix factorization technique known as *Singular Value Decomposition* (SVD).

The weight matrix $\mathbf{W}^L_{m \times n}$ can be approximated by $\hat{\mathbf{W}}^L_{m \times n}$ by taking only the first $k$ singular values of the full matrices [4].

$$\mathbf{W}^L_{m \times n} \approx \hat{\mathbf{W}}^L_{m \times n} \tag{1}$$

$$\hat{\mathbf{W}}^L_{m \times n} = \mathbf{U}_{m \times k} \mathbf{D}_{k \times k} \mathbf{V}_{k \times n} \tag{2}$$

$$\hat{\mathbf{W}}^L_{m \times n} = \mathbf{U}_{m \times k} \mathbf{B}_{k \times n} \tag{3}$$

As shown in Equation 3, the original weight matrix $\mathbf{W}^L_{m \times n}$ can be expressed in a concise form as the product of $\mathbf{U}_{m \times k}$ and $\mathbf{B}_{k \times n}$ by introducing a new layer between $L$ and $L+1$. The updating of the states of all nodes can be expressed as:

$$\hat{\mathbf{W}}^L \cdot \mathbf{x}^L = (\mathbf{U} \cdot \mathbf{B}) \cdot \mathbf{x}^L = \mathbf{U} \cdot (\mathbf{B} \cdot \mathbf{x}^L) \tag{4}$$

Thus, the number of pairwise calculations and weight parameters is reduced when: $k < \frac{m \cdot n}{m+n}$ [6].

### 4.3 Architecture and Hyper-parameter Search

The optimization of the hyper-parameters associated with various RL models are essential to the model performance. The original DQN model created was analyzed in order to identify number of layers, layer types (e.g convolutional, fully-connected) and kernel sizes and thus identify applicable techniques. We adopt a relatively simple guided search based on expected cost reductions in system resources. The aggressiveness of this search is guided by the Retain Rate. With respect to this component the Retain Rate is proportional to the number of parameters retained in an optimized model compared to the original.

### 4.4 Convolution Minimization

Conceptually related to §4.2, this method differs in that it targets convolutions. Just a previously discussed, this method while popular as an approach to optimizing the inference of deep classification models – *little experimental results exist about how these techniques function in a Deep RL module.*

The optimization opportunity comes from the time complexity of convolving multi-channel input data, $\mathbf{x} \in$ $\mathbb{R}^{C \times H \times W}$ with a bank of $N$ $d \times d$ filters denoted as, $\mathcal{K} \in \mathbb{R}^{N \times d \times d \times C}$ is $\mathcal{O}(CNd^2HW)$[29]. Therefore, by exploiting the property of separable filters we are able to reduce the overall convolutional operations[6]. The output feature map $\mathbf{M} \in \mathbb{R}^{N \times H \times W}$ can be stated as:

$$M_j = f\left( \sum_c \mathbf{x}^c * \mathcal{K}_j{}^c + \mathbf{b}_j \right) \tag{5}$$

Where, $f(\cdot)$ is a non-linear function, $\mathbf{b}$ is the bias vector for the layer and $c$ is the index over $C$ input channels. Thus, for convolution minimization the goal is to obtain an approximation $\hat{\mathcal{K}}$, expressed as [6]:

$$\hat{\mathcal{K}}_n^c = \sum_{k=1}^K \mathcal{H}^k{}_n \, (\mathcal{V}^c{}_k)^T \tag{6}$$

Where, $\hat{\mathcal{K}}$ can be decomposed into horizontal, $\mathcal{H} \in \mathbb{R}^{N \times 1 \times d \times K}$ and vertical, $\mathcal{V} \in \mathbb{R}^{K \times d \times 1 \times C}$ filters with lower ranks, controlled by the parameter $K$. The reconstruction error of the convolutional filter approximation should be small. Under this approach, the overall convolution task (indexed by $n$) becomes:

$$\mathcal{K}_n * \mathbf{x} \approx \hat{\mathcal{K}}_n * \mathbf{x} \tag{7}$$

$$\hat{\mathcal{K}}_n * \mathbf{x} = \sum_{c=1}^C \sum_{k=1}^K \mathcal{H}^k{}_n \, (\mathcal{V}^c{}_k)^T * \mathbf{x}^c \tag{8}$$

$$= \sum_{k=1}^K \mathcal{H}^k{}_n * \left( \sum_{c=1}^C \mathcal{V}^c{}_k * \mathbf{x}^c \right) \tag{9}$$

Therefore, filters $\mathcal{V}$ and $\mathcal{H}$ are updated by the approximation $\hat{\mathcal{K}}$ using the *Convolution Minimization* technique from the initial DQN model shown in Figure 2. It can be seen from Equation 9 that the input $\mathbf{x}$ is now convolved with two successive layers with filters, $\mathcal{V}$ and $\mathcal{H}$ instead of the original convolutional layer. Thus, the overall memory consumed by the original convolutional layers can be reduced while maintaining comparable model performance on the embedded platforms.

### 4.5 Optimization Tuner

As the Deep RL model proceeded through a batch of iterations it always considered possible optimization strategies to follow. We make no novel contribution with this optimization process and use a greedy search guided by some regression models that give it guidance towards tuning each optimization method for improved reward score outcomes combined with reduced resource usage. In this system, the *Retain Rate* is utilized as a universal metric in guiding the exploration of optimization opportunities as it was technique independent. This approach provides an effective strategy in identifying good areas of model compression when using various combinations of *Resource Scalers* as well as bad areas in which model compression lead to degraded performance.

# 5. EVALUATION

We now present the main benefits achieved by our framework. The aim of each experiment can be concisely as:

- **Classic RL**: Investigate performance of optimized DQN agents in classic RL control scenarios.
- **Robot RL**: Profile the memory footprint of robot Deep RL models e.g, *Roboschool*.
- **Distributed Coordination**: Show an optimized DQN agent's performance in learning customized message passing protocols for *Colour-Digit MNIST*.

Each experimental setting becomes progressively more advanced and finally tests our framework under an RL that learns a message passing protocol on embedded devices. Our results demonstrate that:

- Optimized DQN models in both the classic and real-world applications have smaller memory footprints when compared to their baseline models.
- A reduction in energy consumption and faster model inference as shown in §5.3.
- Optimized DQN agents in Distributed Coordination with a retain value of 0.65 and 0.32 only had a 0.11 and 0.34 decrease in average reward respectively.

## 5.1 Methodology

The approach taken in this exploration was to identify several simulated and real-world environments for Deep RL development. After which one case study on real embedded devices were performed.

**Metrics.** We will outline two crucial metrics (e.g Retain Rate, Reward Score) used in optimizing Deep RL models and evaluating their performance. The reward score is also known as the cumulative discounted return. It is the reward the DQN agent obtains by interacting with the environment. The reward score is a universal metric used in RL to easily compare the performance of various RL algorithms for applications. *Retain Rate* is presented as a generic term to describe the lowering of complexity. As it is interpreted by different framework optimization components differently direct interpretation is difficult. But during experiments we highlight opportunities for doing this when they arise.

**Hardware Processors.** Most experiments deploy Deep RL to an ARM Cortex M0 processor [22]. These include our Classic RL experiment (CartPole) as well as our final case study. This processor is very limited, and presents a strong test of it Deep RL can be effective on TinyML devices. We also test under a physics-based simulated environment (§5.2.2), and for these experiments use an Amazon Web Services EC2 Instance [3]. But resources were limited artificially, and measured to see how well our framework performed.

## 5.2 Benchmarks: Simulated Environments

We begin our experiments into optimization of Deep RL algorithms with two well known RL simulation environments namely, *CartPole* from OpenAI's *Gym* [7] and *Humanoid* from OpenAI's *Roboschool* [18]. This enables our resource optimized agents to make decisions within an environment and setup open and consistent across the reinforcement learning community.

### 5.2.1 Classic Reinforcement Learning Scenarios

**Experiment Setup.** The DQN agents are run on the ARM Cortex M0.

**Scenario: CartPole.** The game of CartPole is to keep balance of a pendulum and prevent it from falling over by controlling the cart to which the pendulum is attached by an un-actuated joint. The episode fails whenever the pole deviates more than 15 degrees from vertical or the cart moves more than 2.4 units from the center. A reward of 1 is obtained for every frame the pole is kept upright. The total score is the cumulative points it earned during each episode, as the episode has the maximum length of 200 frames, the upper bound of the performance is 200. The initial DQN model was trained for 175k episodes before optimization techniques were applied.

According to Figure 4, we found that the average reward score across 1000 episodes for the retain rate of 0.95 is comparable with that of the original model. There is a decline in the model performance for retain rates ranging from 0.95 to 0.65. Correspondingly, the average reward score decreases from 200 to approximately 120. This is a decrease in reward of 80 across a compression of 30%. Table 1 presents a lower inference time exerted by the compressed (retain=0.95) CartPole model than the original model on the ARM Cortex M0.

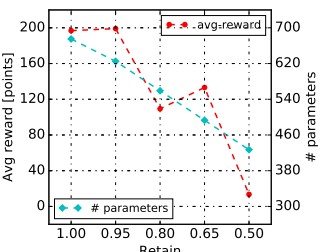

**Figure 3:** CartPole

**Figure 4:** Influence on average reward and parameter number with different retaining levels for *CartPole* when executed on ARM Cortex M0.

**Table 1:** Inference Time of *CartPole* (retain=0.95).

| Model | Time[msec] |
|---|---|
| Original | 2.1 |
| Compressed | 1.2 |

### 5.2.2 Physics-based Robot Environment

The performance and memory footprint of DQN agents applied to one of OpenAI's *Roboschool* environments namely, *Humanoid*. The memory profile associated with each model was captured using Python's Memory Profiler[25] and executed on an AWS instance.

**Experiment Setup.** The DQN agent is constructed based on the architecture outlined in §4 for continuous action spaces.

**Table 2:** Average Reward Score of Humanoid in *Roboschool*.

| Game | Reward Score |
|------|-------------|
| Humanoid | -56.871 |

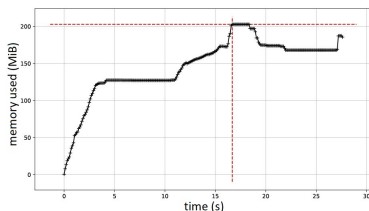

**Figure 5:** Time Series Memory Profile of DQN Agent in *Humanoid* environment.

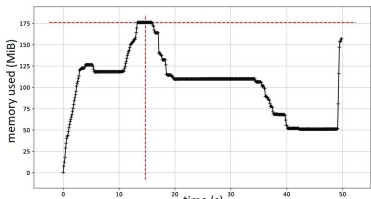

**Figure 6:** Time Series Memory Profile of A3C Agent in *Humanoid* environment (Peak=175.5MB).

**Table 3:** Peak Memory Footprint of DQN Algorithm.

| Game | Memory (MB) |
|------|-------------|
| Humanoid | 232.3 |

**Scenario: Humanoid.** The *Humanoid* environment was utilized to control a bipedal robot with an upper torso to walk forward as fast as possible and for as long as possible without tipping over.

*Reward Score.* Table 2 presents the reward score achieved which is close to the unmodified RL agent.

*Memory Footprint Varies Dependant on RL Algorithm.* The memory profile of the DQN and A3C algorithms shown in Figure 5 and 6 are different. The A3C agent has a sharp increase in memory usage before tapering off while the DQN agent's memory footprint gradually increases. The A3C agent has a lower memory footprint peak of 175.5MB when compared to DQN's 200.6MB.

*Memory Footprint Increases in High Dimensions.* The DQN agent has a peak memory footprint of 232.3MB (See Table 3). The higher memory footprint can also be described as having a higher retain rate which is representative of complexity associated with the *Humanoid* model. The retain rate can be thought of as universal controller.

## 5.3 Case Study: Distributed Coordination

These experiments study the proposed optimization techniques applied to two practical real-world embedded and robot applications: the autonomous robot navigation and distributed coordination network protocol.

### 5.3.1 Distributed Coordination Network Protocol

Reinforced Inter-Agent Learning (RIAL) and Differentiable Inter-Agent Learning (DIAL) techniques have been proposed as means of learning multi-agent Deep RL[12]. RIAL uses deep Q-Learning with recurrent neural networks where agents train individually and share model parameters (*Decentralized Learning*). Whereas in DIAL, gradients of agents are passed through communication channels to each other leading to system that is end-to-end trainable (*Centralized Learning*). The ability of agents to learning efficient and robust message passing protocols across noisy channels is beneficial in reducing the size and number of messages needed through reinforcement learning. This approach can greatly decrease the computation and energy requirements and memory footprint on resource constrained devices. Previous research in this area has mainly been theoretical. Therefore, we propose a practical experiment to demonstrate DIAL agents learning customized message passing on the resource constrained ARM Cortex MO[1]. We also demonstrate the performance of compressed DIAL agents in comparison to the original model.

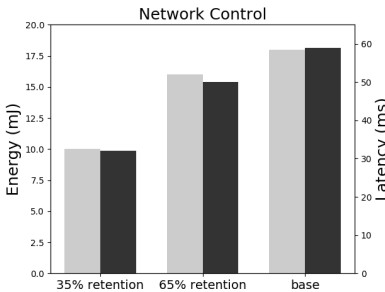

**Figure 7:** Network control performances of three DQNs on snapdragon 400.

**Communication Scenario.** We implement the deep Q-learning solution on the ARM Cortex M0 and adopt the DIAL approach within a game called *Colour-Digit MNIST*[12]. We adapt this agent to learn a more practical message passing protocol: Each sensor node observes a low resolution image of a moving object from which it is able to extract a series of binary characteris-



**Figure 8:** Runtime Protocol Behavior of two Compressed DIAL agents (*"RNN 32"* and *"Weight compression 65"*) compared to the original algorithm. Reward score (y-axis) is a measure of the quality of protocol decisions. This strong result shows both compressed agents are only slightly worse than the original.

tics that include (1) if the object is a car or a different entity; (2) if the object color is red or not; (3) if the object has two or four wheels. We setup a 5-node network, with each node acting as a DIAL agent. Every node captures the three pieces of information from an image and executes a state exchange protocol towards agreeing to each characteristic they observe. Nodes can repeat the analysis on the image if needed. The learning between DIAL agents seeks to find a custom specific message passing protocol for this specific problem, that can cope with channel noise and that will minimize the amount of bits exchanged between nodes. There are two phases, a learning phase in which agents freely exchange information towards the learning of an optimized efficient protocol given conditions and a runtime phase during which the learned protocol is executed.

**Experiment Setup.** Each ARM processor is networked with simulated wireless conditions (random message drop with a probability of 0.42) conditions. Within the simulation each ARM processor is provided with the same image at each protocol execution. *Although network conditions are simulated, the actual necessary computation is performed on an ARM Cortex M0 – this satisfies the core aim of the experiment.*

**Reward Score Metric.** In the context of the following results, the reward score corresponds to the efficiency of the protocol. The objective of our embedded optimization is to reach as closely the performance of the original agent behavior defined by [12]. In understanding these results a 10% lower score at the run-time of the protocol should be roughly treated as 10% less efficient in terms of the number of bits exchanged.

**Optimized Protocol Decisions Remain Accurate.** The model achieved a maximum reward score after training for 5000 episodes as specified by [12]. This experiment shows that the behavior of the optimized protocol, as executed by the DIAL agents, remain very faithful to the unoptimized agents. Figure 8, shows the comparison of the average reward score for the unchanged agent

(far left) and two optimized variations of the agent that allow them to execute efficiently even on an ARM Cortex M0. The two optimizations are a compression of the RNN layers of the agent using all three techniques described in §3 (*"RNN 32"*), while the second (*"Weight compression 65"*) applies only the redundancy compression alone. This translates to 32% and 65% of the original number of agent parameters being retained.

**Significantly Lower Resource Consumption.** Figure 7 illustrates the performances of the control scenario under three retain factors - 35%, 65%, and 100% baseline. It can be seen from the figure that the overall energy benefits of the optimized models are significant.

## 6. RELATED WORK

**Optimizing Discriminative Deep Learning.** Towards improving the performance of *deep*-forms of reinforcement learning specifically for constrained devices, the most directly applicable techniques are those already developed for this purpose within the context of discriminative deep models. The past two years has seen an explosion of research of this type (e.g., [16, 27, 5, 13, 20, 15, 9, 17, 34]). These works provide underlying insights as to *how* potential modifications to the workflow of DRL algorithms and the neural architectures that are used towards trading off the quality of Deep RL decisions for lower resources. But the research we conduct here has developed methods to exploit these same observations within Deep RL and been first to quantify if the degraded quality is still acceptable.

**Deep Reinforcement Learning.** The concept of *optimizing* reinforcement learning even pre-dates the use of deep learning methods and principles. Optimization in reinforcement learning is not a novel idea and several researchers approached this problem since the 2000s.

But only recently relatively recently have works started to examine this important issue of on-device tinyML deployment of DeepRL [11, 33, 19, 28]. We add to this growing discussion by providing a general purpose framework for TinyML applications. [28] targets networking applications and is less general purpose. [19] is particularly relevant to our mixed precision component, and we plan to leverage its results in the future. [33, 11] are strong examples of applications of Deep RL running on TinyML systems as drones. Our case study is distinct from drones, as it targets networking protocols, and complements this strong drone based work.

## 7. CONCLUSION

In this work, we have taken significant strides towards understanding of the system resource bottlenecks within forms of deep reinforcement learning (Deep RL), and how these can be managed with a series of proposed optimizations that make it possible for mobile-class pro-

cessors adopt such approaches.

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
