# OpenReview forum: "Resource Efficient Deep Reinforcement Learning for Acutely Constrained TinyML Devices"
_tinyml.org/tinyML/2021/Research_Symposium — tinyML 2021 Poster_

### Official Review · AnonReviewer2 · 2021-01-28

**Overall Merit Score:** 1

**Brief Summary:**

This paper proposes a set of tools to optimize a deep RL model so that it can run on TinyML platforms with acute resource constraints. The paper applies model compression techniques to deep RL models and builds a system that can run RL tasks like DQN and distributed coordination on ARM Cortex M0 embedded processor.

**Detailed Comments:**

This paper identifies the important problem of running deep RL agents on TinyML devices, and successfully builds working system showing the effectiveness of their model compression method. However, my main concern of the paper is that the main component of the proposed method, the model compression techniques, are not novel and a lot of details are not provided. For example:
1.	For the quantization, is it applied on weight? Activation? Gradient? Or all of them? Is the quantization applied for both training and inference?
2.	How exactly is the compressed model size (i.e. k in redundancy minimization and precision in quantization) determined? Are they kept the same throughout the learning process?
3.	“Retain rate” is mentioned multiple times throughout the paper, yet there’s no explanation on how exactly it is defined and how is it interacting with the model compression techniques
4.	How the “framework architecture” introduced in Sec 3.1 interacts with each other is also not well explained.
Besides these missing details, I do think it would be important to include an ablation study on each individual technique to see how it is contributing to the final performance.

To sum up, I do think the motivation and the result of this paper would be interesting for the TinyML community and may inspire more future works. Yet the missing details and the lack of proofreading make the current paper unsuitable for publication. Therefore, I would suggest rejecting the paper, but encourage the author to present and discuss this work as a poster in the conference.

**Paper Strengths:**

1.	The problem tackled by the paper, running RL agent on resource-constraint devices, is an important problem and worth further investigating
2.	Detailed evaluation on real devices is provided, showing the proposed method can effectively reduce resource consumption without significantly hurting the performance of the RL agent


**Paper Weaknesses:**

1.	All the model compression techniques used in this paper are directly borrowed from computer vision domain. No novel method is proposed in the paper and there’s no analysis on whether these techniques suit the deep RL setting
2.	Some important details in the proposed method are not adequately introduced. See detailed comments for examples
3.	The paper considers three different techniques for model compression, yet there’s no ablation study analyzing whether all these techniques are useful and how much each one of the three techniques contributes to the final compression rate.
4.	This paper needs a thorough proof reading, some sentences (e.g. the first two sentences of section 3.2) may contain grammar errors that make them very hard to understand


**Poster (If Paper Is Rejected):**

1: Yes, ok for poster sesion to nurture work

**Reviewer Confidence:**

4: The reviewer is confident but not absolutely certain that the evaluation is correct

---

### Official Review · AnonReviewer4 · 2021-01-30

**Overall Merit Score:** 2

**Brief Summary:**

This paper identifies the system resource bottlenecks for deploying reinforcement learning systems on mobile platform. Several general-purpose techniques are proposed to reduce overall system complexity and make it possible to run RL algorithms on mobile processors.

**Detailed Comments:**

1. Use line chart instead of bar chart in Figure 7. Figure 8 can be replaced with a table.

2. Make separate evaluations on each proposed technique for better demonstrations of their efficiencies.


**Paper Strengths:**

The paper proposed to use several techniques for reducing the computation complexity and memory usage of standard reinforcement learning algorithms. Experiments are deployed on mobile processor to demonstrate the efficiency of proposed method.

**Paper Weaknesses:**

1. Some contents need to be clarified. How retain rate is determined? In 4.3, what is the definition of cost reductions in system resources. What are the steps to apply all the proposed techniques? A flow chart or pseudo-code will help.

2. Evaluation metrics and results are not sufficient to demonstrate the efficiency of proposed method. For CartPole experiment, it is expected to provide details of computation complexity (e.g., number of operations, FLOPs, etc.). For Humanoid experiment, what are the reward scores and memory usages for the original models.

3. Details of each network model used in experiments are not provided.



**Poster (If Paper Is Rejected):**

1: Yes, ok for poster sesion to nurture work

**Reviewer Confidence:**

4: The reviewer is confident but not absolutely certain that the evaluation is correct

---

### Official Review · AnonReviewer1 · 2021-01-30

**Overall Merit Score:** 2

**Brief Summary:**

In this work, the authors proposed a framework for reducing the resource overhead for deep reinforcement learning by utilizing optimized agent algorithms and neural network architectures to make it deployable on Tiny ML devices. The authors study a variety of general-purpose techniques (memory, computation and energy reduction requirements) designed to lower system resource bottlenecks for Deep RL. The proposed framework includes selective mixed parameter precision, architecture search, redundancy minimization as well as convolution optimization. The architecture consists of Deep RL module, along with resource scaler and optimization tuner. Method is evaluated under Open AI Cart pool and Humanoid simulators. Also, authors present a Deep RL application running on ARM Cortex M0 as an example of a distributed co-ordination task execution on tiny ML devices.

**Detailed Comments:**

I provided detailed comments in the pervious section. While the manuscript demonstrates great engineering and integration efforts, I believe its scientific contribution is limited.

**Paper Strengths:**

1.	This is a well-motivated paper with good technical depth and experimental results.
2.	The approach of optimizing deep RL networks to reduce memory footprint is a relatively new avenue and will open up new research directions for deployment of deep RL networks into resource constrained devices.
3.	Distributed coordination case study presents a unique learning structure where agents pass customized messages with in them to reduce resource requirements.


**Paper Weaknesses:**

1.	The proposed framework consists of 4 units that deploy neural network architecture search, quantizer, redundancy minimization using SVD, and separable convolution layers. None of these techniques are author’s novel work. They are all controlled through a unit called optimization tuner. The optimization module is vague and there is not much detail provided about it.
2.	Despite mentioning the backpropagation and network update, there is a little explanation about it in the text. The way that simulation environments are deployed on ARM-Cortext M0 is also unclear.
3.	the variable retain rate is very loosely explored in the paper as the authors claim that this is a generic term signifying reduction of complexity. But how this computational complexity is a factor of the retain and if the reduction is linear needs answering.
4.	Results are inadequate. more detailed analysis must be provided provide. For example, the Table1 presents results for only one case. The measurement technique for that case is unclear too.
5.	Cart Pole and Humanoid are very simple RL environment. How does the framework perform in more complex environments?
6.	The alternative agents and their relation to the retain rate needs further exploration.
6.   There is no actual comparison with similar works.
7.      There are several writing errors in the text and figures. Some of which are:
·         Why the related works is section 6? It must be after the introduction
·         There are numerous usages of Italic fonts for sentences in the text. They must be removed and make the text uniform.
·         Fig 1 and Fig 2 is not good representation of Reinforcement Learning concept and it must be revised. Input of the neural net must be on the left side (not top) and output on the right side (not down). States are fed to the network and the actions come out of it.
·         Figure 3 needs to be organized. X-axis numbers must be written from small to large not Vice versa.
·         Figures 4 is missing or two captions are written for figure 3.
·         Some of the figures doesn’t have legend.
·         Figures must be referenced sequentially in the text. Figure 8 is addressed before figure7
·         Why the reward for the humanoid is negative?


**Poster (If Paper Is Rejected):**

1: Yes, ok for poster sesion to nurture work

**Reviewer Confidence:**

5: The reviewer is absolutely certain that the evaluation is correct and very familiar with the relevant literature

---

### Official Review · AnonReviewer3 · 2021-01-30

**Overall Merit Score:** 1

**Brief Summary:**

The authors propose to optimize Deep Reinforcement Learning algorithms for resources-limited environment. An optimjization framework and some benchmarking are shown throughout the paper.

**Detailed Comments:**

see strengths and weaknesses box above

**Paper Strengths:**

- Deep Reinforcement Learning techniques are interesting to address at TinyML and could be used by many different applications.
- Authors correctly listed previous works in the domain

**Paper Weaknesses:**

- Overall, the paper is really not convincing for TinyML. It looks like the authors rephrased the paper to fit to TinyML but a lot of evaluations are not valid. According to me, drones and robots use a lot of power and resources.
- The proposed framework is not really a framework and this is more a list of optimization that are well-known. Some of them, like Optimization Tuner, lack of explanations and will not bring a lot of value to the readers
- My major concern is about the "framework" evaluation:
-- ARMM0: this is absolutely not clear which resources are used here. The most important resource to optimize in order ot save power is memory and the authors report a memory footprint between 175MB and 200MB. This is a huge number and clearly not representing any kind of low power applications.
-- in the end, some benefits are shown but the overall envelope is far from being tiny


**Poster (If Paper Is Rejected):**

1: No, paper is below bar for poster as well

**Reviewer Confidence:**

4: The reviewer is confident but not absolutely certain that the evaluation is correct

---

### Decision · Program_Chairs · 2021-02-05

**Decision:**

Accept (Poster)

**Comment:**

Thank you for your submission.

Following careful consideration by our reviewers, we regret to inform you that we are unable to accept your submission.

Please refer to the reviewer comments for your reference. We hope you find this information helpful for submission to another venue, and we hope to see more of your work in the future.